# Exploring and Engineering Novel Strong Promoters for High-Level Protein Expression in *Bacillus subtilis* DB104 through Transcriptome Analysis

**DOI:** 10.3390/microorganisms11122929

**Published:** 2023-12-06

**Authors:** Ji-Su Jun, Hyang-Eun Jeong, Kwang-Won Hong

**Affiliations:** Department of Food Science and Biotechnology, College of Life Science and Biotechnology, Dongguk University, Goyang-si 10326, Republic of Korea; jiiiisoo@naver.com (J.-S.J.); clove2602@naver.com (H.-E.J.)

**Keywords:** *Bacillus subtilis* DB104, protein expression system, promoter engineering, transcriptome-based analysis, human epidermal growth factor

## Abstract

*Bacillus subtilis* is widely employed for recombinant protein expression. *B. subtilis* DB104 offers a distinct advantage as a protein expression host because it is an extracellular protease-deficient derivative of *B. subtilis* 168. We have conducted a time-course transcriptome analysis of *B. subtilis* DB104 in a prior study. In the present study, we identified 10 genes that exhibited strong expression at each time point or all, based on transcriptome data. Subsequently, we assessed the strength of 12 promoters that transcribe these genes using enhanced green fluorescent protein (eGFP) as a reporter. Among these promoters, P_sdp_ and P_skfA_ had the highest expression levels. At 24 h, these two promoters exhibited 34.5- and 38.8-fold higher strength, respectively, than the strength of P43, the control promoter. Consequently, these two promoters were selected for further development. We enhanced these promoters by optimizing spacer length, promoter sequence, Shine–Dalgarno sequence, regulator binding sites, and terminator sequences. As a result, we successfully engineered the most potent protein expression cassette, P_sdp_-4, which exhibited a 3.84-fold increase in strength compared to the original P_sdp_ promoter. Furthermore, we constructed an expression cassette for a human epidermal growth factor (hEGF) using P_sdp_-4 to evaluate its general application. The expression level of His tagged hEGF, quantified using ImageJ analysis and applied to SDS-PAGE, reached the highest yield of 103.9 μg/mL under the control of P_sdp_-4 at 24 h. The expressed hEGF protein was purified, and its bioactivity was confirmed through a cell proliferation assay using HT-29 cells. Our work demonstrates the construction of a highly efficient expression system for *B. subtilis* DB104 based on transcriptome data and promoter engineering. This system enables rapid, inducer-free protein expression within 24 h. It can be used as a valuable tool for various industrial applications.

## 1. Introduction

Recombinant protein products using bacteria hold significant industrial, pharmaceutical, and research values. *Escherichia coli* is the most commonly used host strain for recombinant protein production. It has been successfully used to produce various proteins including collagen peptide [1], cellulose [2], endoglucanase [3], and fucosidase [4]. However, *E. coli* comes with certain disadvantages, such as a propensity of expressed proteins to aggregate [5,6], its production of endotoxin [7,8], and plasmid instability [9]. On the other hand, *Bacillus subtilis*, a Gram-positive bacterium, is known for not producing endotoxins with well-characterized genetic properties. It has garnered attention as a reliable workhorse due to its ‘generally recognized as safe’ (GRAS) status [10]. Over the years, extensive genetic engineering and expression systems have been developed in *B. subtilis* to produce recombinant proteins [11,12,13].

In protein expression systems, two major types of promoters are widely used: inducible promoters and constitutive promoters. Several inducible promoters have been developed and employed in *B. subtilis*, using inducers such as isopropyl β-D-1-thiogalactopyranoside (IPTG) [14,15]. However, the cost of inducers can be prohibitive for industrial applications. On the other hand, genes under the control of constitutive promoters can be expressed throughout all growth phases without requiring specific stimuli for expression.

Developing a highly efficient expression cassette, including robust promoters, stands out among critical strategies for constructing an expression system. Various studies have been conducted regarding screen promoters, including the use of endogenous promoters [16,17,18,19] such as housekeeping gene promoters [20,21] and the construction of synthetic promoter libraries [22,23,24]. However, these approaches are time-consuming and labor-intensive. Notably, most promoters in synthetic promoter libraries often yield suboptimal results [25,26]. Recently, transcriptomic analysis has emerged as a valuable tool for identifying strong promoters that drive high-level gene expression, producing many proteins. This technique increases production efficiency, ultimately leading to time and cost savings. Recent researchers have excavated strong promoters based on transcriptome data in various organisms, including *B. licheniformis* [27], streptomycetes [28], *Pichia pastoris* [29], and *Aspergillus oryzae* [30]. Furthermore, using transcriptomic data, Liu et al. [31] discovered a potent promoter in *B. subtilis*. In that study, they identified the highly expressive promoters P_sodA_ and P_ydzA_, which exhibited higher expression levels than P43, a well-recognized promoter, as strong promoters in *B. subtilis* [32,33].

Moreover, promoters can be engineered to enhance expression. Promoter engineering techniques involve modifying core regions, TATA box sequences, and regulator binding sites [34,35,36], resulting in a stronger promoter with a suitable affinity for the RNA polymerase or regulator binding site. For instance, Lee et al. [37] achieved a five-fold increase in *lacZ* expression by changing −35 and −10 boxes of the *cry3Aa* promoter. Similarly, the *srfA* promoter was improved by optimizing the −35 region sequence, resulting in a 360% increase in the expression of aminopeptidase and a 50% increase in the expression of nattokinase [38].

At the onset of sporulation, *B. subtilis* cells exhibit a social behavior called cannibalism to delay sporulation [39]. Key players in this system are genes belonging to two cannibalism toxin operons: the sporulation delay protein (*sdp*) operon and the sporulating killing factor (*skf*) operon. These genes are transcribed by P_sdp_ and P_skfA_, respectively, engineered in this study. The lysed cells, resulting from the killing factor, provide nutrients for siblings to sustain growth [40].

Epidermal growth factor (EGF) is a protein renowned for stimulating cell growth and proliferation by binding to its receptor, EGFR [41,42]. Given its crucial role in wound healing [43], EGF has been a target for recombinant expression, leading to numerous studies on recombinant EGF expression. For example, Hu et al. [44] reported the production of fibroblast EGF using an IPTG-inducible promoter in *B. subtilis*, with a yield of 31 mg/L. Although a nisin-inducible expression system in *Lactococcus lactis* could reach a productivity of 2.67 mg/L [45], it requires the addition of 10 ng/mL nisin to induce human EGF (hEGF) production.

In our previous study, a time-course transcriptome analysis of *B. subtilis* DB104 was conducted to determine the gene’s expression patterns during growth and to identify highly expressed genes at different time points or overall [46]. In this study, we attempted to develop a novel recombinant protein expression system by selecting the promoters with the highest expression during the growth cycle of *B. subtilis* and engineering these promoters. The activities of screened promoters were assessed using enhanced green fluorescent protein (eGFP) as a reporter [47]. Then, two highly active promoters, P_sdp_ and P_skfA_, were engineered to develop strong protein expression cassettes. Ultimately, we successfully developed constitutive expression cassettes P_sdp_-4 and P_skfA_-2. Notably, P_sdp_-4 enabled the production of bioactive hEGF protein with function, highlighting its utility as a novel tool for recombinant protein expression.

## 2. Materials and Methods

### 2.1. Bacterial Strains, Growth Conditions, and Transformation

The bacterial strains and plasmids used in this study are shown in Table 1. *E. coli* DH5α was used as the cloning host. *B. subtilis* DB104 was used for genomic DNA donor and recombinant expression. These strains were routinely cultured in Luria-Bertani (LB) medium (10 g/L tryptone, 10 g/L NaCl, 5 g/L yeast extract, and pH 7.5) at 37 °C. If needed, the LB medium was supplemented with 50 μg/mL ampicillin for *E. coli* or 10 μg/mL kanamycin for *B. subtilis*. Recombinant *B. subtilis* DB104 cells were grown in LB medium at 30 °C under aeration, with shaking at 250 rpm, for protein expression. The transformation of *E. coli* DH5α was performed using the heat shock method [48]. *B. subtilis* DB104 was transformed as described previously [49].

### 2.2. Analysis of Transcriptome Data and Construction of Expression Vectors

To select strong promoters, previous transcriptome profiling data during the growth of *B. subtilis* DB104 were used [46]. Ten gene promoters, highly expressed at each time point or at all, were selected, based on transcripts per million (TPM) value (Figure 1). The expression pattern of the selected gene was validated by real-time quantitative reverse transcription polymerase chain reaction (RT-qPCR). Primers used for RT-qPCR are listed in Appendix A. *B. subtilis* DB104 was cultivated and sampled as previously described [46]. RNA was extracted from six samples taken at different time points (8, 10, 12, 15, 18, and 24 h) using RNAiso (Takara, Otsu, Japan) following the manufacturer’s instructions. cDNA was synthesized using random hexamer (Roche, Base, Switzerland) and M-MLV reverse transcriptase (Promega, Madison, WI, USA), following the manufacturer’s protocols. RT-qPCR was conducted using Taq Pro Universal SYBR qPCR Master Mix (Vazyme, Nanjing, China) on a CFX Connect Real-Time System (Bio-rad, Hercules, CA, USA). To normalize the relative expression level of target gene, *rpsJ* was used as an internal reference gene.

The strengths of promoters were analyzed using eGFP as a reporter protein. Primers used for plasmid construction are listed in Appendix A. A series of promoter–reporter expression vectors containing each promoter upstream of the *egfp* gene were constructed. In addition, the eGFP expression vector under the control of P43 promoter (promoter of *cdd*) was constructed and used as a control for a comparative study of promoter strength. Promoter sequences of each gene were amplified by PCR using PrimeSTAR Max DNA polymerase (Takara Bio Inc., Shiga, Japan) with corresponding primers. Genomic DNA of *B. subtilis* DB104 was used as a template for promoter sequence amplification. The *egfp* gene was amplified from pUB19-Psin2-*egfp* [34] using corresponding forward primers and common reverse primer (eGFP-R). The vector skeleton was generated by the digestion of the same plasmid with *Mlu*I-*Hind*III. The PCR products of each promoter and *egfp* were recombined by overlap PCR. The PCR-amplified fragments were also digested with *Mlu*I-*Hind*III and ligated with vector skeleton. All promoters constructed were confirmed by sequencing.

### 2.3. Engineering of Expression Cassettes

Recombinant plasmids, containing mutated P_sdp_ and P_skfA_ expression cassettes, were constructed using appropriate primer sets, as detailed in Appendix A. Primers Psdp-F and PskfA-F, initially used for constructing pUB19-P_sdp_-egfp and pUB19-P_skfA_-egfp, respectively, were recycled for the engineering process. Mutant promoters and *egfp* genes, amplified using proper primer sets and template plasmids, were combined by overlap PCR, resulting in the generation of DNA fragments P_sdp_-1, 2, 4, a, b, c, and d, and P_skfA_-1, a, b, and c. Each DNA fragment was digested with *Mlu*I-*Hind*III and subcloned into the vector skeleton. Terminator fragment used for the construction of P_sdp_-3 was amplified using the primer set sdp-F3/sdp-R3. Similarly, terminator fragments used for the construction of P_skfA_-2 and P_skfA_-3 were amplified using skfA-ter-F/skfA-ter-R. They were then digested with *Not*I-*Hind*III and subcloned into the previous plasmids that had been digested with the same restriction enzyme, generating expression cassettes with a terminator downstream of the *egfp* structure gene.

### 2.4. Measurement of Fluorescence Intensity of Enhanced Green Fluorescent Protein (eGFP)

A single colony of *B. subtilis* DB104 recombinants carrying an appropriate expression vector was inoculated into 10 mL of LB broth containing 10 μg/mL kanamycin and incubated at 30 °C, with shaking at 250 rpm. The preculture was transferred into 50 mL of fresh LB, containing the same antibiotics, in a 500 mL baffled flask (initial OD_600_ = 0.1). During culturing for 48 h, samples were taken at indicated time points. Cell density was measured at OD_600_. One hundred μg of the sample was loaded into a black flat 96-well plate (SPL Life Sciences, Pocheon, Republic of Korea). The fluorescence intensity of eGFP was measured with a Victor™ X4 multi-plate reader (PerkinElmer, Waltham, MA, USA). Data were averaged from three independent experiments.

### 2.5. Human Epidermal Growth Factor (hEGF) Expression Using Developed Expression Cassette

The *hegf* gene shown in Appendix A was synthesized by Bioneer Co. (Bioneer Co., Daejeon, Republic of Korea) and amplified using primer set hegf-OF/hegf-R (Appendix A). The His-tag sequence was amplified from pET-15b using the primer pair sdp-his-OF/his-hegf-OR and connected with *hegf* gene by overlap-PCR, resulting in the generation of DNA fragment his-hegf. The *egfp* gene of expression cassette P_sdp_-4 was replaced with *hegf* gene, using overlap-PCR of DNA fragment his-hegf and promoter fragment amplified with the primer pair Psdp-F/sdp-his-R. The resulting amplicon, P_sdp_-his-hegf, was digested with *Mlu*I-*Hind*III and subcloned into the vector skeleton. Consequently, a recombinant plasmid pUB19-P_sdp_-hegf with the His-tag sequence inserted between the promoter and the *hegf* gene was generated.

### 2.6. Purification of His-Tagged Human Epidermal Growth Factor (hEGF) and Thrombin Treatment

A single colony of *B. subtilis* DB104 strain harboring pUB19-P_sdp_-hegf was inoculated with LB broth containing 10 μg/mL kanamycin and incubated at 30 °C with shaking at 250 rpm. The preculture was then transferred into a 500 mL baffled flask, containing 50 mL of fresh LB medium with the same antibiotics (initial OD_600_ = 0.1). During cultivation, the culture was sampled at 12 h and 24 h. Samples were subjected to purification and SDS-PAGE. Quantification of protein bands was conducted using ImageJ software (National Institutes of Health, Bethesda, MD, USA, Version 1.53m). A MagListo His-tagged protein purification kit (Bioneer, Daejeon, Republic of Korea) was used to purify His-tagged hEGF protein from the supernatant sample at 24 h, following the manufacturer’s instruction. Prior to elution, thrombin was used to obtain His-tag-removed hEGF protein. Cleavage of the His-tag was performed by adding 6 U of thrombin per 1 mL of the sample in a cleavage buffer (50 mM Tris-HCl, pH 8.0, 10 mM CaCl_2_) at room temperature overnight. Two types of samples with and without thrombin cleavage are hereafter referred to as His-hEGF and -hEGF, respectively. The remaining His-tag was also eluted to confirm the efficiency of thrombin treatment and referred to as His-.

### 2.7. Enzyme-Linked Immunosorbent Assay (ELISA)

The three samples, His-hEGF, -hEGF, and His-, were detected using a Human EGF ELISA kit (Assay Genie, Dublin, Ireland) and a His-tag Protein ELISA kit (Abcam, Cambridge, MA, USA), according to each manufacturer’s instructions.

### 2.8. Cell Proliferation

The human intestinal epithelial cell line, HT-29 cells (ATCC, Manassas, VA, USA), were maintained in Dulbecco’s modified Eagle medium (DMEM, Welgene, Seoul, Republic of Korea) supplemented with 10% fetal bovine serum (FBS, Gibco, Bulington, Canada) and 1% penicillin/streptomycin (Gibco, Crand Island, NY, USA). The cells were cultured at 37 °C in a humidified atmosphere containing 5% CO_2_. Cell proliferation was measured by MTT (3-(4,5-Dimethylthiazol-2-yl)-2,5-diphenyltetrazolium bromide) assay. Briefly, HT-29 cells (2 × 10^5^ in 200 μL per well) were seeded into a 96-well plate. After 24 h of incubation, the medium was replaced with an equal volume of DMEM containing either 10 ng/mL EGF standard or -hEGF and incubated for 48 h. Subsequently, 110 μL of the medium was removed, and 10 μL of MTT was added to each well, followed by incubation for 3 h. The medium was then removed, and 100 μL dimethyl sulfoxide (DMSO) was added to each well followed by incubation for another 30 min at the same condition. The absorbance was measured at 590 nm using an ELISA reader. Cell proliferation was compared to that of the control (cells cultured in DMEM alone). The results were generated from three independent experiments.

### 2.9. Plasmid Stability

A plasmid stability test was performed, following the methodology outlined by Li et al. [51], to determine the durability of the recombinant plasmid throughout protein production. Briefly, a single colony of the recombinant *B. subtilis* DB104 strain was inoculated into LB broth containing 10 μg/mL of kanamycin. It was then cultured at 30 °C with continuous agitation at 250 rpm for 48 h. At 12 h intervals, samples were spread onto LB agar plates with or without kanamycin (10 μg/mL). The plasmid retention rate was calculated using the following formula: plasmid retention rate (%) = (number of colonies on LB plate with kanamycin/number of colonies on LB plate without kanamycin) * 100. Each experiment was conducted with three replicates.

### 2.10. Statistical Analysis

All data are presented as average ± standard deviation from independent triplicate experiments. The statistical significance was determined by Student’s *t*-test or one-way analysis of variance (ANOVA), followed by the Duncan’s post hoc test using IBM SPSS Statistic 27.0 software (SPSS Inc., Chicago, IL, USA). Differences with *p* < 0.05 were considered statistically significant.

## 3. Results

### 3.1. Selection of Strong Promoter Based on Transcriptome Analysis Data

In our previous study, we conducted a transcriptome analysis of *B. subtilis* DB104 during the whole growth stage: the middle of the exponential growth phase (8 h), the end of the exponential growth phase (10 h), the beginning of sporulation (12 h), the former part of sporulation (15 h), the latter part of sporulation (18 h), and the end of sporulation (24 h) [46]. We screened genes with the highest TPM value from the RNA-seq data to find strong promoters. The *hag* gene (flagellin protein) and *sdpC* gene (killing factor) were the two genes with the most expression in the middle of the exponential growth phase (8 h). *sdpC* also had the highest expression level at the end of the exponential growth phase (10 h). Another killing factor gene, *skfA*, and spore coat morphogenetic protein gene, *spoIVA*, showed the highest expression levels at the beginning of sporulation (12 h). The small acid-soluble spore protein genes *sspE* and *sspB* showed the highest expression at the formal part of sporulation (15 h). At the latter part of sporulation (18 h), the two topmost expressive genes were spore coat protein genes *cotX* and *cotY*. The putative toxin genes *yczN* and *yczM*, which belong to one operon, had the highest TPM at the end of sporulation (24 h). Consequently, six genes, *sdpC*, *skfA*, *sspE*, *cotX*, *cotY*, and *yczN*, belonged to the top seven genes with the highest expression. Therefore, a total of 10 genes (*hag*, *sdpC*, *skfA*, *spoIVA*, *sspE*, *sspB*, *cotX*, *cotY*, *yczN*, and *yczM*) most highly expressed at each time point or throughout growth were selected (Figure 1 and Appendix A).

### 3.2. The Effect of Selected Promoters on the Expression of Enhanced Green Fluorescent Protein (eGFP)

Before estimating promoter strength, the transcription levels of the genes transcribed by promoter candidates and control gene *cdd* were verified by RT-qPCR (Appendix A). Promoter sequences were predicted using the DBTBS database [52], while the upstream sequence (about 350 bp) from the start codon was used for the *yczNM* operon, whose promoter sequence was not identified. To assess the promoter strengths of 10 selected genes, their promoter sequences were amplified by PCR and connected with the reporter protein gene, *egfp*, by overlap PCR. Among them, promoters of genes with more than one promoter, *sdpC*, *spoIVA*, and *cotX*, were divided as shown in Figure 2B–D. In the *sdp* operon, to which *sdpC* belongs, there are two promoters in front of the first gene, *sdpA*, and the third gene, *sdpC* [53,54], and they were assessed separately (P_sdp_ and P_sdpC_). In the case of *spoIVA*, there are two promoters in a row. They were evaluated together or only the second promoter separately as P_spoIVA-1_ and P_spoIVA-2_, respectively. The *cotX* gene belongs to *cotVWX* operon, consisting of three structure genes (*cotV*, *cotW*, and *cotX*) and two distinct promoters. These two promoters (P_cotX_ and P_cotVWX_) were also assessed separately. The pUB19-Psin2-*egfp* plasmid construct in a previous study [34] was used to make an *E. coli*–*Bacillus* shuttle vector backbone (Figure 2A). Therefore, a total of 12 expression vectors harboring each promoter candidate (P_hag_, P_sdp_, P_sdpC_, P_skfA_, P_spoIVA-1_, P_spoIVA-2_, P_sspE_, P_sspB_, P_cotX_, P_cotVWX_, P_cotYZ_, and P_yczNM_, Table 1) were generated.

*B. subtilis* DB104 recombinants hosting different promoters were grown in LB broth and the relative fluorescence unit (RFU) driving expression of *egfp* was measured at 24 h and 48 h (Figure 3A,D). The results showed that promoter activities of P_sdp_, P_sdpC_, P_skfA_, P_spoIVA-1_, P_cotVWX_, P_cotX_, and P_cotYZ_ were significantly higher than those of control promoter P43 at both time points. In the case of P_hag_ and P_yczNM_, their activity had relatively lower significance at 24 h and 48 h, respectively. It was found that eGFP expression levels by P_spoIVA-2_, P_sspE_, and P_sspB_ were not significantly different from that by P43. As shown in Figure 3C, *B. subtilis* DB104 recombinants with promoters other than P_sdp_ had a similar pattern of entering the stationary phase from 12 h or 18 h. However, the amount by which cell density decreased after that varied by strain. The strain containing P_sdp_, which had the most unique growth pattern, had the highest OD_600_ value, and exponential growth continued until the latest time of 30 h. The two strains harboring P_sdp_ and P_skfA_ exhibited the highest RFU, being 34.5- and 37.8-fold higher at 24 h compared to the control strain, respectively. The expression levels of eGFP of these strains were measured in shorter time intervals (Figure 3D,E). Their expression patterns were quite different. The strain harboring P_sdp_ reached its maximum value, with a rapid increase in RFU, at 18 h. However, its value decreased afterward. Although the RFU also increased sharply at 18 h, the value of the strain with P_skfA_ steadily increased after that, showing the highest value at 48 h, which was 19-fold higher than that of the control promoter P43. Therefore, we sought to develop a strong protein expression system, by engineering these two promoters that demonstrated the highest eGFP expression.

### 3.3. Expression Cassette Engineering

#### 3.3.1. Engineering of P_sdp_ Expression Cassette

To improve the P_sdp_ expression cassette, some elements were modified as described in Table 2 and Figure 4. First, the spacer between the start codon and SD sequence was optimized. To reduce the number of nucleotides, 27 bp immediately following the SD sequence or directly preceding the start codon were deleted (designated as P_sdp_-1 or P_sdp_-a, respectively, Figure 4A). In the strain containing P_sdp_-1 and P_sdp_-a, 280% and 9% fluorescence were expressed, compared to the strain with P_sdp_ at 48 h, respectively (Figure 4B and Appendix A). Although a spacer length of 8 bp was applied equally to these two strains, the results were the opposite. Subsequently, the SD sequence of P_sdp_-1, AGAGGAGG, was replaced with a strong SD sequence, TAAGGAGG, resulting in an enhanced fluorescence level of 180% compared to the original promoter at 48 h (P_sdp_-2, Figure 4 and Appendix A). By inserting a terminator sequence of the *sdp* operon downstream of the *egfp* structural gene in cassette P_sdp_-2, it was observed that the expression of eGFP increased by 38.3-fold compared to the strain with the original promoter at 48 h (P_sdp_-3, Figure 4). The P_sdp_ cassette has −35 and −10 boxes candidates assumed to be a σ^H^- dependent promoter. Their sequence did not perfectly match with the consensus sequence of the −35 and −10 boxes of the σ^H^-dependent promoter in *B. subtilis*. Therefore, the −35 and −10 boxes of P_sdp_-3 cassette were changed to the consensus sequence of the σ^H^-dependent promoter in *B. subtilis* (P_sdp_-b, Figure 4A). However, the maximum RFU value of P_sdp_-b decreased to 72%, compared to that of P_sdp_-3 (Appendix A). Meanwhile, other −35 and −10 boxes candidates were assumed to be a σ^A^-dependent promoter. The sequence of −10 box matched with the consensus sequence of σ^A^-dependent promoter in *B. subtilis*, but the −35 box did not match. The −35 box was also changed to a consensus sequence. However, the resultant expression cassette, P_sdp_-c, only exhibited 34% of the eGFP expression level of P_sdp_-3 at 48 h (Figure 4A and Appendix A)**.** The AbrB and Rok bind to the *sdp* promoter and act as transcription inhibitors. To improve the P_sdp_-3 expression cassette, the binding sites of AbrB and Rok, located near the Spo0A binding site and −35 boxes, were simultaneously removed, leading to a 127 bp deleted expression cassette, P_sdp_-d (Figure 4A). Contrary to expectations, the eGFP expression was decreased by a factor of 0.38, compared to that of P_sdp_-3 at 48 h (Appendix A). The expression cassette P_sdp_-4 was also generated from P_sdp_-3 by replacing the repressor AbrB binding site with the activator Spo0A binding site (Figure 4A). There was no significant difference in maximum RFU between P_sdp_-3 and P_sdp_-4. However, P_sdp_-4 exhibited significantly higher expression levels at 24 h, achieving faster expression than P_sdp_-3 by 12 h.

#### 3.3.2. Engineering of P_skfA_ Expression Cassette

To enhance the P_skfA_ expression cassette, several modifications were made, as described in Table 3 and Figure 5. The σ^A^-dependent promoter P_skfA_ originally contained the nucleotide sequence TATTAT at its −10 box position, which did not match the consensus sequence of that of the σ^A^-dependent promoter in *B. subtilis*. Therefore, it was changed to the consensus sequence (P_skfA_-1, Figure 5A). This change resulted in the highest RFU value expressed by cassette P_skfA_-1. The RFU value increased 1.36-fold, compared to the original P_skfA_ cassette. Additionally, by inserting the terminator sequence of the *skf* operon downstream of the *egfp* structural gene in the original cassette P_skfA_, the expression of eGFP was increased 1.69-fold, compared to the original cassette at 48 h (P_skfA_-2, Figure 5). However, combining these two modifications, changing in −10 box and inserting the terminator, resulted in poor eGFP expression (P_skfA_-3, Figure 5B). Furthermore, other elements of the original promoter P_skfA_ were modified. The SD sequence of P_skfA_, AGAGGAGG, was replaced with the strong SD sequence, leading to a reduced fluorescence level of 69% compared to the original promoter at 48 h (P_skfA_-a, Figure 5A and Appendix A). The consensus sequence of AbrB binding site, which represses *skfA* transcription, was changed from TGGTA to TGATA (P_skfA_-b, Figure 5A and Appendix A). The maximum RFU value of P_skfA_-b was decreased to 58%, compared to that of P_skfA_. Finally, the consensus spacer length between −35 and −10 boxes was adjusted to that of the original P_skfA_ promoter, generating P_skfA_-c expression cassette (Figure 5A and Appendix A). The eGFP expression by P_skfA_-c was only 32% of that of P_skfA_ at 48 h.

### 3.4. Human Epidermal Growth Factor (hEGF) Expression Using Developed Expression Cassette

The developed expression cassette, P_sdp_-4, utilizing eGFP as the reporter protein, exhibited the highest expression level, which was 342% higher than the P43 control promoter at 24 h. A heterologous protein, hEGF, was then chosen to evaluate its broad applicability. An expression vector, pUB19-Psdp-hegf, was constructed and introduced into *B. subtilis* DB104. Figure 6A shows the expression box for hEGF using P_sdp_-4. The expression of His-tagged hEGF protein was evaluated through SDS-PAGE and ImageJ analysis (Figure 6B and Appendix A). The results showed the successful expression of His-tagged hEGF protein in *B. subtilis* DB104 under the control of P_sdp_-4, with a yield of 103.9 μg/mL at 24 h. As described in the Materials and Methods section, the sample taken 24 h after incubation was purified, resulting in His-hEGF, -hEGF, and His-. When using an ELISA kit for human EGF, His-hEGF was not detected, while -hEGF was detected (Figure 6C). When using an ELISA kit for His-tag protein, a subequal amount of His-tag protein was detected in the His-hEGF and -hEGF samples (Figure 6D). Furthermore, there were rare His-tag proteins in the -hEGF sample. These results indicated that recombinant hEGF protein was successfully separated from His-tag protein.

The cell proliferation test was then conducted on HT-29 cells to assess the bioactivity of the produced protein, -hEGF. After 2 days, cell proliferation was significantly increased by both EGF standard and -hEGF (Figure 7). These data suggest that the hEGF protein derived from the developed expression cassette, P_sdp_-4, exhibited biological activity similar to that of the standard product.

### 3.5. Stability of Recombinant Plasmids

During the 48 h of protein production, it was confirmed that the plasmid remained stable (Figure 8). In the case of strain harboring pUB19-P_sdp_-egfp-4, the plasmid retention rate was kept at 100% until the end of incubation. However, in the case of the strain harboring pUB19-P_sdp_-hegf, the plasmid retention rate decreased by about 63% from 24 h.

## 4. Discussion

A fundamental approach to controlling protein expression involves regulating transcript production at the promoter level. In this study, we analyzed previous transcriptome data [46] to screen potential strong promoters. Therefore, ten genes with high TPM values were selected (Figure 1 and Appendix A). They are involved in spore formation in this organism, except for *hag*. This result suggests that sporulation-related genes are highly expressed throughout growth. Nicolas et al. [54] first reported that *sdpC* could be transcribed from more than one promoter. Our results here confirmed that the upstream of the gene could be a promoter region. However, our results indicated that the gene’s high TPM value resulted from P_sdp_, the first promoter of the operon. Along with *sdpC*, promoters of *spoIVA* and *cotX* were employed in a divided manner (Figure 2B–D). As a result, 12 promoters were used to construct expression vectors, and their strengths were evaluated by measuring eGFP expression levels.

Despite the selection of all promoters based on high transcription levels, there was not a perfect correlation between the levels of mRNA expression (TPM value) and protein expression (RFU value) (Appendix A). For example, eGFP expression driven by P_sspE_ and P_cotX_ promoters were low, while total TPM values of the gene were high (Appendix A). Interestingly, their expression patterns corresponded to each other, except for a slight delay in eGFP expression. The TPM values of *skfA* and *spoIVA* peaked at 12 h. The promoter activities of these genes peaked at 15 h. Similarly, the activity of P_hag_ promoter peaked at 12 h, where the TPM value of *hag* reached its highest level at 12 h. Despite their high transcript level, these results indicate that the promoter activities should be verified before biotechnological application. Our suggestion is consistent with result of Miao et al. [55].

Because the two promoters, P_sdp_ and P_skfA_, exhibited the highest fluorescence units at 24 h and 48 h after incubation (Figure 3A,D), they were used for further engineering. In *B. subtilis*, the spacer length between SD sequence and the start codon is known to affect protein yield [56,57]. To apply the optimal spacer length to P_sdp_, which had a longer spacer length at 35 bp, 27 nucleotides before the start codon (P_sdp_-1) or after the SD sequence (P_sdp_-a) were deleted. The application of an 8 bp spacer length yielded contrasting results (Figure 4B and Appendix A). To date, the length of the spacer has been considered crucial for gene expression. However, our results showed that the sequence of the spacer is also significant. The ribosome binding site sequence, the 5′ untranslated region (UTR), can influence gene expression. The translation initiation efficiency in procaryote depends on the strength of the binding interaction between the SD sequence on the mRNA and the complementary anti-SD sequence located at the 3′ end of the 16S rRNA [58,59]. Accordingly, a significantly increased expression level of eGFP was observed when the strong SD sequence of *B. subtilis* was applied to P_sdp_-1 (P_sdp_-2, Figure 4B). However, the fluorescence intensity decreased when the same sequence was applied to P_skfA_ (P_skfA_-a, Appendix A). In other words, modifying the SD sequence could positively or negatively affect protein expression. Although it is difficult to explain the reason for these results because the dataset is limited in size, it can be concluded that the optimal sequence of SD depends on other factors within the promoter sequence.

The 3′ UTR region terminator is a crucial element for gene expression. It plays roles in allowing the polymerase to stop RNA elongation and enhancing RNA stability [60,61]. We introduced the terminator sequence from *sdp* operon and *skfA* operon to the P_sdp_-2 and P_skfA_ expression cassettes to increase protein yield, resulting in P_sdp_-3 and P_skfA_-2, respectively. As a result, the eGFP expression was significantly increased in both cases (Figure 4 and Figure 5). Curran et al. [62] demonstrated that the terminator-influenced transcript levels increased 6.5-fold for high-strength promoters, and 11-fold for low-strength promoters in *Saccharomyces cerevisiae*. However, when we applied the same terminator to the developed promoter P_skfA_-1, fluorescent expression nearly disappeared below the levels of the original promoter. These results underscore the significance of proper transcription termination in enhancing gene expression. At the same time, our results demonstrate that both the promoter and terminator regulate gene expression and their interaction can have a complex effect on gene expression levels.

Numerous other studies have confirmed the dependence of promoter strength on the core promoter sequence [17,63,64,65]. Our previous study [34] successfully increased the *sinIR* promoter strength by modifying the −35 and −10 region by approximately 40%. Specifically, the −10 box sequence of P_skfA_, a σ^A^-dependent promoter, was altered from TATATT to TATAAT. Only a single-nucleotide variation resulted in a 36% increase in the maximum RFU value (Figure 5). In the case of P_sdp_, the exact −35 and −10 box sequences of the promoter are unknown. However, referring to the study of Strauch et al. [66], showing that the operon has a σ^A^-dependent promoter, we modified two nucleotides of the −35 region sequence to match the consensus sequences (P_sdp_-c). Against expectation, the RFU value dropped to 34%, compared to that before the modification (Appendix A). Subsequently, we attempted another approach, using a σ^H^-dependent promoter sequence instead of a σ^A^-dependent promoter. It is known that σ^H^ regulates the gene activated at the early stationary phase [67], where the *sdpC* gene is highly expressed in our transcriptome data ([46], Appendix A). Two regions within P_sdp_ resemble the −35 and −10 boxes of the σ^H^-dependent promoter. One base in each box was changed to match the consensus sequence (P_sdp_-b). However, similar to P_sdp_-c, it showed a 72% decrease in fluorescence expression (Appendix A). These results suggest that σ^H^ and σ^A^ might not control the *sdp* operon. Another possible reason is that modified nucleotides might be at a site where other regulators involved in gene expression are bound.

Transcription factors (TFs) bind to the DNA sequence and regulate transcription in either a positive manner (as an activator) or a negative manner (as a repressor). The *sdp* operon is controlled by three TFs: AbrB, Rok, and Spo0A [66,68,69,70]. It is known that both AbrB and Rok bind to the promoter of *sdp* operon and act as repressors, although their precise binding sites remain unknown. Therefore, we deleted two broad AbrB binding sites previously reported by Strauch et al. [66] and the consensus Rok binding site in the middle, totaling 127 bp, to weaken the effect of the repressor on P_sdp_. Unfortunately, the resulting expression cassette, P_sdp_-d, showed a 62% reduction in fluorescence expression compared to that before the deletion. Thus, we replaced the repressor binding site with the activator Spo0A binding site, as a different strategy. Interestingly, replacing the AbrB binding site with the Spo0A binding site did not increase the maximum fluorescence value compared to that before replacement. However, it accelerated expression, resulting in a 46% increase in expression at 24 h. Engineered expression cassettes, namely P_sdp_-4 and P_skfA_-2, were developed through a series of processes. Their strengths were much stronger than the strength of the promoter of *aprE*, which was previously identified as highly expressive by Miao et al. [55], with the maximum fluorescence intensities of P_sdp_-4 and P_skfA_-2 being 8.3 and 6.3 times higher, respectively (Appendix A).

Furthermore, to explore their applicability in expressing different proteins, we constructed an expression cassette using P_sdp_-4 to express hEGF (Figure 6A). To facilitate easy purification of the recombinant protein, we introduced a His-tag and a thrombin cleavage site in front of the structure gene. As a result, 103.9 μg/mL of His-hEGF, in which the His-tag remained intact, was produced. A previous study by Hu et al. [44] produced human fibroblast EGF using the IPTG induction system in *B. subtilis*. This result suggests that the developed constitutive P_sdp_-4 expression cassette can produce more proteins than IPTG induction. Furthermore, their study increased protein production through fed-batch culture by approximately 2.7-fold [44]. These approaches can be implemented in future research using our developed expression cassette.

The plasmid retention rate of pUB19-P_sdp_-hegf began to decrease at 12 h, while pUB19-P_sdp_-egfp-4 remained stable until 48 h (Figure 8). These results indicate that plasmid stability is influenced by the gene they harbor. This problem could be resolved by integrating expression cassette into the chromosome. For example, Cha et al. [71] enhanced the genetic stability of glucoamylase by integrating it into *S. cerevisiae*. In addition, -hEGF could be successfully separated from the His-tag via thrombin treatment. It is important to highlight that His-hEGF, in which the His-tag remained intact, was undetectable through human EGF ELISA (Figure 6C). This result indicates that the N-terminal His-tag can impact the structure of recombinant hEGF, disturbing its binding to the ELISA antibody. This can be restored through thrombin cleavage. Consistent with this result, Western blot did not detect N-terminus His-tagged hEGF, while C-terminus His-tagged hEGF was well-detected when produced in *Nicotiana benthamiana* [72]. Furthermore, the recombinant hEGF demonstrated bioactivity, as the proliferation level of cells expressing the recombinant hEG was similar to that of the control EGF protein in a cell proliferation test using HT-29 cells.

## 5. Conclusions

In conclusion, this study developed two highly potent protein expression cassettes, P_sdp_-4 and P_skfA_-2, through a transcriptome-based approach and promoter engineering. Notably, P_sdp_-4 successfully overexpressed hEGF with proper bioactivity. Therefore, the high-level expression system, using P_sdp_-4 and P_skfA_-2, could be a valuable tool for expressing recombinant industrial enzymes, hormones, and pharmaceutical proteins without needing an induction process.

## Figures and Tables

**Figure 1 microorganisms-11-02929-f001:**
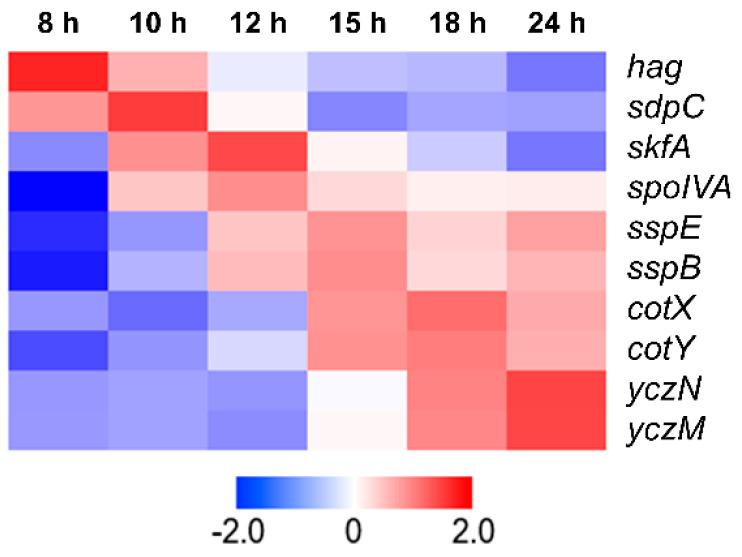
Heat map of transcriptome analysis for 10 selected genes from *Bacillus subtilis* DB104. The heat map shows Z-scores for highly expressed genes by time, based on transcripts per million value. The gene names are shown on the right. The color from red to blue indicates high to low expression levels.

**Figure 2 microorganisms-11-02929-f002:**
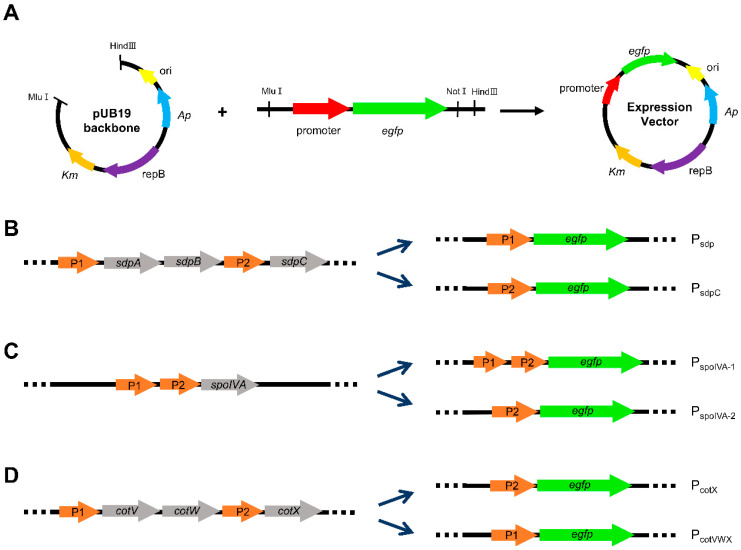
The construction scheme for expression vector containing different promoter candidates. (**A**) Overview of the expression vector used for promoter strength assessment. *Ap* and *Km* signify resistance markers for ampicillin and kanamycin, respectively. Meanwhile, ori and repB stand for the replication origin and replication protein B, respectively. Diagrams for the *egfp* fusion construction for promoters of (**B**) *sdp* operon (**C**) *spoIVA* and (**D**) *cotVWX* operon, which have more than one promoter.

**Figure 3 microorganisms-11-02929-f003:**
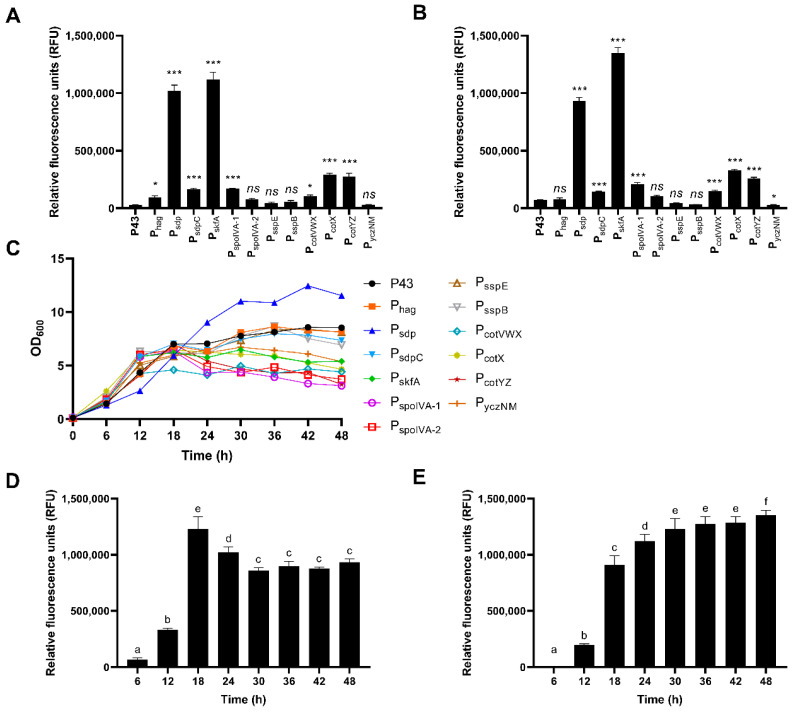
Evaluation of highly active promoters. The eGFP expression level under the control of promoter candidates at (**A**) 24 h and (**B**) 48 h. In (**A**,**B**), the significant levels are indicated by “*”, “***”, and “*ns*” for *p* < 0.05, *p* < 0.001, and *p* > 0.05, compared to P43, respectively (Student’s *t*-test). (**C**) Comparison of cell growth pattern of the *B. subtilis* DB104 recombinants hosting different promoters. Time-course analysis of fluorescent intensities of two recombinants that host (**D**) P_sdp_ and (**E**) P_skfA_. In (**D**,**E**), means with the same letter are not significantly different from each other (*p* > 0.05 ANOVA followed by Duncan’s post hoc test).

**Figure 4 microorganisms-11-02929-f004:**
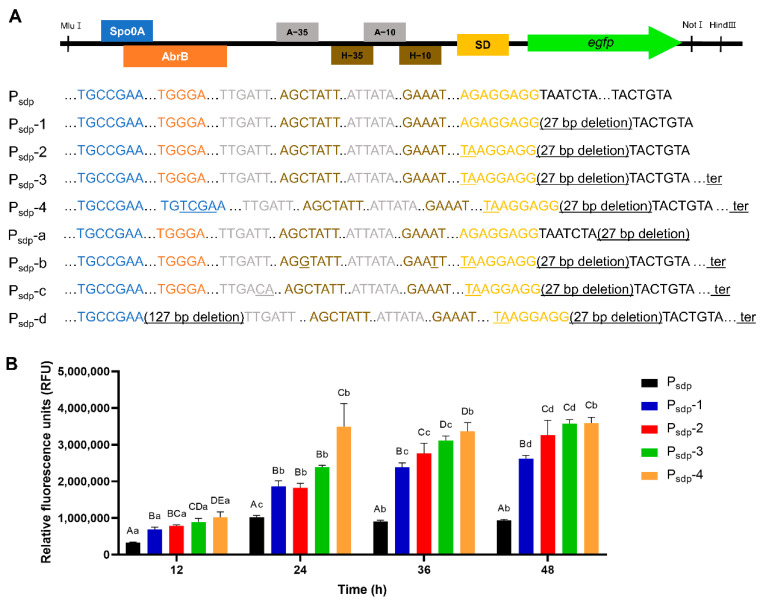
Optimization of P_sdp_ expression cassette. (**A**) Architecture of P_sdp_ expression cassette and the sequence of its derivatives. Underlines indicate changes. ter represents the terminator sequence from *B. subtilis* DB104 *sdp* operon. (**B**) The eGFP expression level of strains containing P_sdp_ expression cassette derivatives. In (**B**), different lower-case letters denote significant differences between different time points within each recombinant strain, while different upper-case letters denote significant differences between expression cassettes at each time point (*p* < 0.05, ANOVA followed by Duncan’s post hoc test).

**Figure 5 microorganisms-11-02929-f005:**
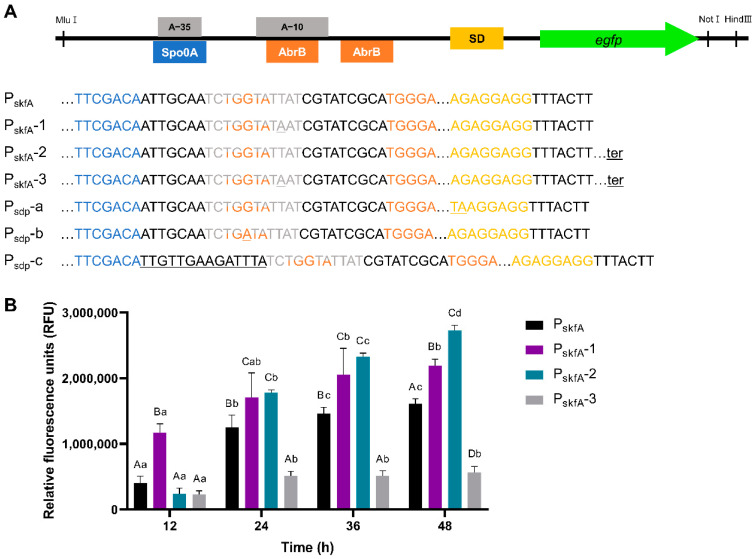
Optimization of P_skfA_ expression cassette. (**A**) Architecture of P_skfA_ expression cassette and the sequence of its derivatives. Underlines indicate changes. “ter” represents the terminator sequence from right behind of *B. subtilis* DB104 *skfA* gene. (**B**) The eGFP expression level of strains containing P_skfA_ expression cassette derivatives. In (**B**), different lower-case letters denote significant differences between different time points within each recombinant strain, while different upper-case letters denote significant differences between expression cassettes at each time point (*p* < 0.05, ANOVA followed by Duncan’s post hoc test).

**Figure 6 microorganisms-11-02929-f006:**
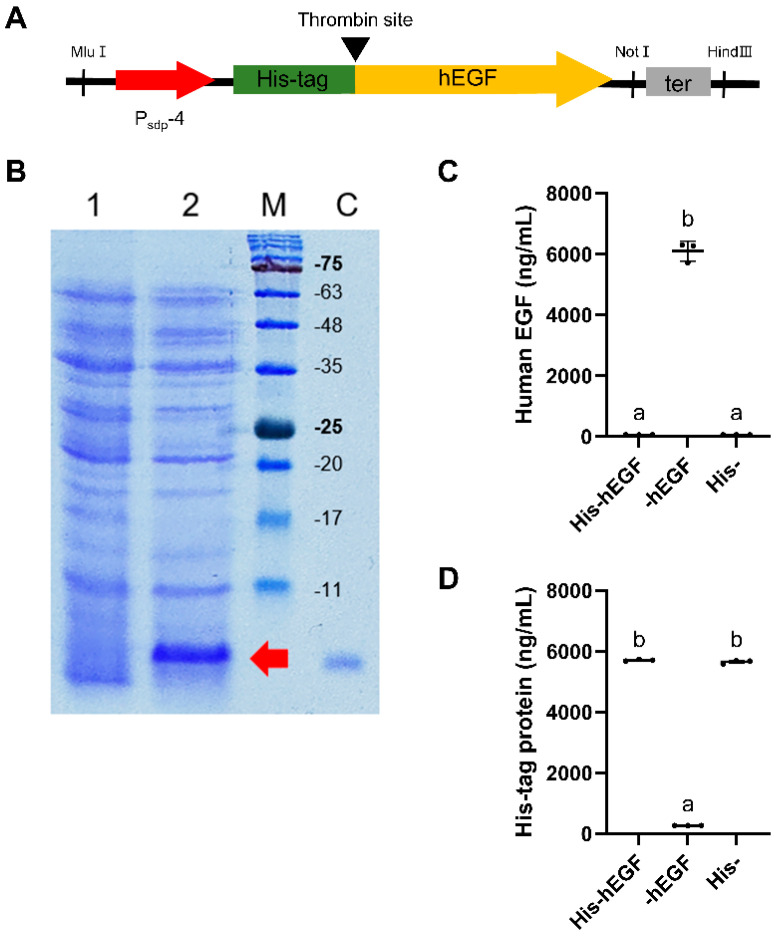
The effect of P_sdp_-4 expression cassette on hEGF protein expression. (**A**) Architecture of the hEGF expression box under the control of the P_sdp_-4 expression cassette. (**B**) Detection of His-tagged hEGF protein expression by SDS-PAGE at 24 h. Lane M, protein molecular maker; lane C, standard hEGF protein (0.5 μg); lane 1, culture samples of *B. subtilis* DB104 with an empty vector pUB19 at 24 h; lane 2, culture samples of *B. subtilis* DB104 carrying pUB19-P_sdp_-hegf at 24 h. His-tagged hEGF bands are indicated by red arrows. His-hEGF concentration was analyzed using ImageJ software. (**C**) Detection of recombinant protein before or after purification by ELISA for human EGF. (**D**) Detection of recombinant protein before or after purification by ELISA for His-tag. In (**C**,**D**), means with the same letter are not significantly different from each other (*p* > 0.05, ANOVA followed by Duncan’s post hoc test).

**Figure 7 microorganisms-11-02929-f007:**
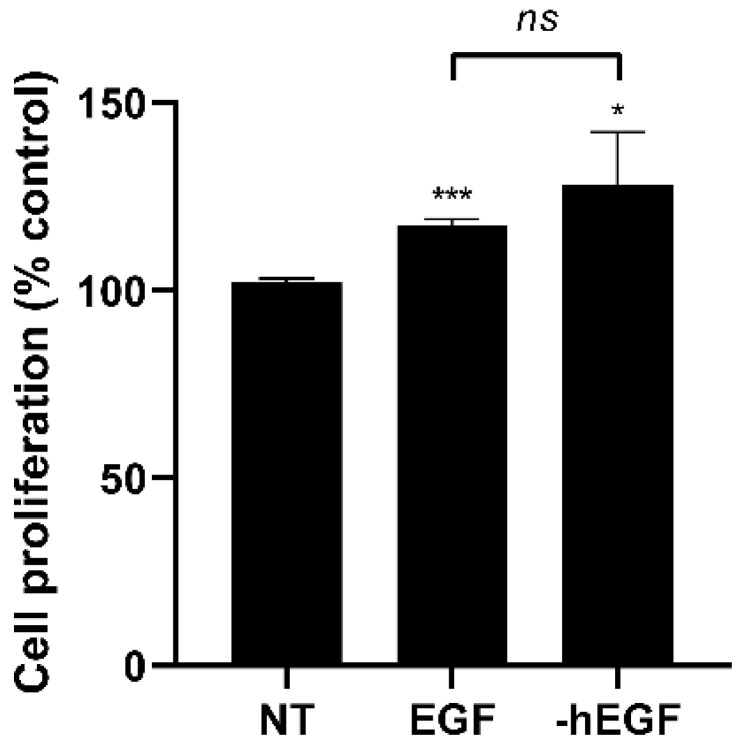
Cell proliferation activity of recombinant hEGF. HT-29 cells were untreated (NT) or treated with EGF standard (EGF) or recombinant hEGF (-hEGF) for 48 h. The significant levels are indicated by “*”, “***”, and “*ns*” for *p* < 0.05, *p* < 0.001, and *p* > 0.05 compared to NT, respectively (Student’s *t*-test).

**Figure 8 microorganisms-11-02929-f008:**
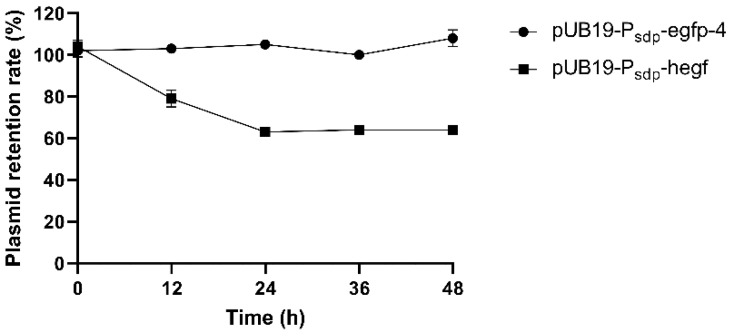
Stability of recombinant plasmid during protein production.

**Table 1 microorganisms-11-02929-t001:** Bacterial strains and plasmids used in this study.

Strains and Plasmids	Description	Resource
Strains		
*Escherichia coli* DH5α	F^−^ Φ80*lac*ZΔM15 Δ(*lac*ZYA-*arg*F) U169 deoR *rec*A1 *end*A1 *hsd*R17 (rK^−^, mK^+^) *pho*A *sup*E44 λ^−^ *thi*-1 *gyr*A96 *rel*A1	Lab stock
*Bacillus subtilis* DB104	*his npr*R2 *npr*E18 Δ*aprA3*	Kawamura and Doi [50]
Plasmids		
pUB19-Psin2-*egfp*	Ap ^r^, Km ^r^, *E. coli*-*Bacillus* shuttle vector pUB19 derivative, *egfp*	Lab stock
pET-15b	Ap ^r^, *E. coli* expression vector	Lab stock
pUB19-P43-egfp	pUB19 with P43 in front of *egfp*	This study
pUB19-P_hag_-egfp	pUB19 with P_hag_ in front of *egfp*	This study
pUB19-P_sdp_-egfp	pUB19 with P_sdp_ in front of *egfp*	This study
pUB19-P_sdpC_-egfp	pUB19 with P_sdpC_ in front of *egfp*	This study
pUB19-P_skfA_-egfp	pUB19 with P_skfA_ in front of *egfp*	This study
pUB19-P_spoIVA-1_-egfp	pUB19 with P_spoIVA-1_ in front of *egfp*	This study
pUB19-P_spoIVA-2_-egfp	pUB19 with P_spoIVA-2_ in front of *egfp*	This study
pUB19-P_sspE_-egfp	pUB19 with P_sspE_ in front of *egfp*	This study
pUB19-P_sspB_-egfp	pUB19 with P_sspB_ in front of *egfp*	This study
pUB19-P_cotX_-egfp	pUB19 with P_cotX_ in front of *egfp*	This study
pUB19-P_cotVWX_-egfp	pUB19 with P_cotVWX_ in front of *egfp*	This study
pUB19-P_cotYZ_-egfp	pUB19 with P_cotYZ_ in front of *egfp*	This study
pUB19-P_yczNM_-egfp	pUB19 with P_yczNM_ in front of *egfp*	This study
pUB19-P_sdp_-egfp-1	pUB19-P_sdp_-egfp derivative, P_sdp_-1 expression cassette	This study
pUB19-P_sdp_-egfp-2	pUB19-P_sdp_-egfp-1 derivative, P_sdp_-2 expression cassette	This study
pUB19-P_sdp_-egfp-3	pUB19-P_sdp_-egfp-2 derivative, P_sdp_-3 expression cassette	This study
pUB19-P_sdp_-egfp-4	pUB19-P_sdp_-egfp-3 derivative, P_sdp_-4 expression cassette	This study
pUB19-P_sdp_-egfp-a	pUB19-P_sdp_-egfp derivative, P_sdp_-a expression cassette	This study
pUB19-P_sdp_-egfp-b	pUB19-P_sdp_-egfp-3 derivative, P_sdp_-b expression cassette	This study
pUB19-P_sdp_-egfp-c	pUB19-P_sdp_-egfp-3 derivative, P_sdp_-c expression cassette	This study
pUB19-P_sdp_-egfp-d	pUB19-P_sdp_-egfp-3 derivative, P_sdp_-d expression cassette	This study
pUB19-P_skfA_-egfp-1	pUB19-P_skfA_-egfp derivative, P_skfA_-1 expression cassette	This study
pUB19-P_skfA_-egfp-2	pUB19-P_skfA_-egfp derivative, P_skfA_-2 expression cassette	This study
pUB19-P_skfA_-egfp-3	pUB19-P_skfA_-egfp-2 derivative, P_skfA_-3 expression cassette	This study
pUB19-P_skfA_-egfp-a	pUB19-P_skfA_-egfp derivative, P_skfA_-a expression cassette	This study
pUB19-P_skfA_-egfp-b	pUB19-P_skfA_-egfp derivative, P_skfA_-b expression cassette	This study
pUB19-P_skfA_-egfp-c	pUB19-P_skfA_-egfp derivative, P_skfA_-c expression cassette	This study
pUB19-P_sdp_-hegf	pUB19-P_sdp_-egfp-4 derivative, *hegf* instead of *egfp*	This study

Ap: Ampicillin; Km: Kanamycin; ^r^: resistance.

**Table 2 microorganisms-11-02929-t002:** Summary of P_sdp_ expression cassette engineering.

Expression Cassette	Description	Engineered From
P_sdp_-1	Optimizing of spacer between start codon and SD (35 bp to 8 bp)	P_sdp_
P_sdp_-2	Application of consensus SD sequence	P_sdp_-1
P_sdp_-3	Terminator insertion after structure gene	P_sdp_-2
P_sdp_-4	Substitution of AbrB binding site to Spo0A	P_sdp_-3
P_sdp_-a	Optimizing of spacer between start codon and SD (35 bp to 8 bp)	P_sdp_
P_sdp_-b	Application of consensus sequence of σ^H^-dependent promoter	P_sdp_-3
P_sdp_-c	Application of consensus sequence of σ^A^-dependent promoter	P_sdp_-3
P_sdp_-d	AbrB binding site deletion	P_sdp_-3

**Table 3 microorganisms-11-02929-t003:** Summary of P_skfA_ expression cassette engineering.

Expression Cassette	Description	Engineered From
P_skfA_-1	Application of consensus sequence of σ^A^-dependent promoter	P_skfA_
P_skfA_-2	Terminator insertion after structure gene	P_skfA_
P_skfA_-3	Terminator insertion after structure gene	P_skfA_-2
P_skfA_-a	Application of consensus SD sequence	P_skfA_
P_skfA_-b	Modification of AbrB binding site to its reverse sequence	P_skfA_
P_skfA_-c	Optimizing of spacer between −10 and −35 (7 bp to 14 bp)	P_skfA_

## Data Availability

The data presented in this study are available on request from the corresponding author.

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
