# Peer review of "Exploring and Engineering Novel Strong Promoters for High-Level Protein Expression in Bacillus subtilis DB104 through Transcriptome Analysis"

_microorganisms, 2023, doi:10.3390/microorganisms11122929_

Round 1

Reviewer 1 Report

Comments and Suggestions for Authors

In this study, the authors screened and engineered the native promoters from Bacillus subtilis, and verified the application in overproduction of recombinant protein thereof. Although the results might be interested to the readers, there are some critical issues should be thoroughly addressed. 

The major question is that how the authors determine the sequences of the selected 10 genes from RNA-seq data? Maybe none of them has been characterized before. Besides, what sigma factors recognize these promoters? There are insufficient data for these questions in my opinion.

In Figure 1, the 10 promoters selected in the article do not maintain high transcriptional levels at all stages. Are these 10 promoters representative? What is the reference for the so-called high transcription level?

The purpose of the article is to screen and obtain high expression of recombinant proteins through engineering modification. However, when selecting from transcriptome, only transcriptional activity was used as the selection criterion, and the impact of mRNA transcription stability on the translation process was not considered. Is it comprehensive to only select mRNA content as the selection criterion?The amount of data is limited, and there is insufficient research on interval deletion in the engineering transformation of promoters.

There is no in-depth exploration of the relationship between interval length and the final expression level of recombinant proteins.

Reviewer 2 Report

Comments and Suggestions for Authors

This is a well-designed and valuable study, the major result of which is the development of high-level expression system to be attractive for production of enzymes and pharmaceutical proteins. All the experimental data and procedures have clearly been presented. Unfortunately, the major weakness of this ms is insufficient introduction of this work and too long discussion. I’d recommend revising this ms and making it more understandable for broad audience than in its current form

Comments

Introduction is rather descriptive and insufficient to justify undertaking of this study.

The leitmotif for this study and novel tasks to be solved have not clearly been presented in Introduction although something is mentioned in Abstract.  The authors published the paper [46 in refs list], and the major finding should be a link to motivate this new study.

Paragraphs LL68-82 should more be tied more to the major idea and goal than in the current ms; yet they look illogical.

“Strong promoters” should be explained for readers.

Obviously,  the sentence “However, it is worth noting that the present study represents the first successful development of promoters…” must be omitted from Introduction. This is the result.

Next, the authors have not explained in Introduction why the sdp operon and skf operons are  beneficial.

The objective of this study should be written and elaborated instead “Herein, we used ….” (LL83-9)

The authors have not mentioned the collection or provider of the strain B. subtilis DB104. Lab stock (Table 1) is not enough

What is “formal part of sporulation”? What is meant by the end of sporulation (24 h)? How was sporulation monitored?

I suspect that the term “ the end of sporulation” has roughly been used here; this means that mature spores liberated from lysed mother cells. Rather, almost all cells started to sporulate in 24 h.  How was the sporulation monitored? I suggest making  revisions for terms (latter part of sporulation, end of sporulation)

Results from this study must not be overlapped with those reported in the previous authors’ study [ref 46]; indeed, I have already met some same numerical data (Table 2, this ms) and the formerly presented.

L122: reference is needed.

Used, using, were used should be avoided in the same sentences.

Revise “During culture for 48 h, the culture (culture…, culture)

Please add “phase” after “the middle of exponential growth”

Typesetting should be checked.

Comments on the Quality of English Language

English is good, few points should be checked. 

Round 2

Reviewer 1 Report

Comments and Suggestions for Authors

In the revised manuscript, the authors have addressed all the questions I had brought forward. It could be accepted.